# Prediction of Sea Surface Temperature in the South China Sea Based on Deep Learning

Peng Hao , Shuang Li *, Jinbao Song and Yu Gao

Institute of Physical Oceanography and Remote Sensing, Ocean College, Zhejiang University, Zhoushan 316021, China
* Correspondence: lshuang@zju.edu.cn

**Abstract:** Sea surface temperature is an important physical parameter in marine research. Accurate prediction of sea surface temperature is important for coping with climate change, marine ecological protection, and marine economic development. In this study, the SST prediction performance of ConvLSTM and ST-ConvLSTM with different input lengths, prediction lengths, and hidden sizes is investigated. The experimental results show that: (1) The input length has an impact on the prediction results of SST, but it does not mean that the longer the input length, the better the prediction performance. ConvLSTM and ST-ConvLSTM have the best prediction performance when the input length is set to 1, and the prediction performance gradually decreases as the input length increases. (2) Prediction length affects SST prediction. As the prediction length increases, the prediction performance gradually decreases. When other parameters are kept constant and only the prediction length is changed, the ConvLSTM gets the best result when the prediction length is set to 2, and the ST-ConvLSTM gets the best result when the prediction length is set to 1. (3) The setting of the hidden size has a great influence on the prediction ability of the sea surface temperature, but the hidden size cannot be set blindly. For ST-ConvLSTM, although the prediction performance of SST is better when the hidden size is set to 128 than when it is set to 64, the consequent computational cost increases by about 50%, and the performance only improves by about 10%.

**Keywords:** sea surface temperature prediction; ConvLSTM; ST-ConvLSTM; deep learning; South China Sea

## 1. Introduction

Sea surface temperature (SST) is an important physical quantity to research and understand the ocean [1–8]. The change in SST is closely related to air–sea interaction and climate change. In addition, the temporal and spatial changes of SST also have a significant impact on the distribution of natural fisheries, artificial aquaculture, and red tide outbreaks, which in turn can affect the entire marine ecosystem. It can be seen that accurate prediction of ocean temperature, especially SST, is of great significance to the research of air–sea interaction, the change of the marine ecosystem, and the sustainable development of the marine economy.

The South China Sea is located in the tropical and subtropical regions in the southern part of the Asian continent, connecting the Pacific Ocean and the Indian Ocean through the Bashi Strait, the Sulu Sea, and the Strait of Malacca. It is characterized by a remarkable tropical maritime climate, with short springs and autumns, long summers, no ice and snow in winters, mild seasons, humid air, and abundant rainfall [9–11]. Especially in the central and southern sea areas, there are high temperatures and high humidity all year round. The seawater temperature is suitable, the water quality is fertile, and the feed is sufficient. It is a feeding and wintering ground for economic fish, and the fishery resources are abundant [12–16]. In addition, the South China Sea is an important component of the western Pacific warm pool, where the air–sea interaction is very strong. The changes in the

western Pacific warm pool have an extremely important impact on the local climate and social and economic development [17–19].

In recent years, the methods of SST prediction have become more and more accurate [20–34]. The methods can be generally divided into three categories: one is the empirical prediction method, which can make a qualitative or quantitative prediction according to the persistence, periodicity, similarity, and correlation with other factors of SST changes. The second is the statistical method, which selects some effective influence factors of the SST field through correlation analysis and uses the mathematical statistics method to predict. In terms of statistical methods currently in use, some methods of multivariate analysis are widely used, such as regression analysis, discriminant analysis, cluster analysis, principal component analysis, similarity analysis, etc. The third is the numerical simulation method. The prediction model is established through the dynamics and thermal equations, and the prediction is made based on a numerical solution according to the given initial and boundary conditions. Among them, the first two methods need to be combined with knowledge of ocean dynamics, and researchers need to have a solid theoretical foundation to improve the accuracy. The last approach often requires large computing equipment to perform complex and detailed simulations of the physical equations in the model.

Essentially, single-point SST forecasting is a temporal prediction problem that takes past time series as input and outputs a fixed number (usually greater than 1) of future time series. Recent advances in deep learning, especially the emergence of recurrent neural networks (RNN), long short-term memory (LSTM), and gated recurrent unit network (GRU), have provided some useful insights into how to solve single-point time series prediction problems [35–46]. The deep learning method extracts feature information by training a deep neural network, which has a stronger feature expression ability. However, in the regional SST prediction problem, there are two key aspects: spatial correlation and temporal dynamics. Although the above three methods can be used to solve the spatiotemporal sequence forecasting problem, they do not consider spatial correlation. Based on the above considerations, the researchers proposed ConvLSTM [47], a combination of a convolutional neural network and a recurrent neural network, and derived an improved model, ST-ConvLSTM [48] based on it. Due to its excellent spatiotemporal prediction performance, it has received extensive attention from experts and scholars in the field of SST prediction research [49–53].

How to design the model structure to get the best SST prediction results? With different layer settings, input length, prediction length settings, etc., the results may be very different. Understanding the impact of different model parameter settings on the SST predictive ability is crucial to accurately predict SST. This study explores the impact of different parameter settings on the performance of the ConvLSTM and ST-ConvLSTM models in predicting SST. By setting different input lengths, prediction lengths, and the number of hidden nodes in the network, we can comprehensively measure the influence of different methods and parameters on the predictive ability of SST.

In Section 2, we describe some preparatory work. In Section 3, we describe the study area, study data, study methods, etc. In Section 4, we give the experimental setting and procedure. In Section 5, we give the experimental results and discuss them in detail. Finally, in Section 6, we summarize our findings and provide an outlook for future work.

## 2. Preliminaries

### 2.1. SST Prediction Using Deep Learning

For the prediction of SST in a certain region, it is essentially a spatiotemporal series prediction problem that takes past time series data as input and a certain amount of future time series data as output. Suppose we need to predict SST over a spatial region represented by $M \times N$ cells consisting of $M$ rows and $N$ columns, where each cell in the grid can map $P$ physical features.

As shown in Figure 1, the data value of a grid point at any time can be represented by a tensor $X \in \mathbb{R}^{P \times M \times N}$. From the perspective of a time dimension, the observations at time

length $t$ form a tensor sequence $X_1, X_2, \ldots, X_t$. Therefore, the SST prediction problem can be defined as a tensor sequence of $J$ time lengths in the past, to predict the tensor sequence of the next $K$ time lengths:

$$\hat{X}_{t+1}, \ldots, \hat{X}_{t+K} = \underset{X_{t+1}, \ldots, X_{t+K}}{arg\ max}\ p\left(X_{t+1}, \ldots, X_{t+K} | X_{t-J+1}, \ldots, X_t\right) \tag{1}$$

SST is one of the most important parameters in the global ocean–atmosphere system. Accurately predicting the temporal and spatial distribution of SST is of great significance for coping with climate change, disaster prevention and mitigation, and marine ecological protection. In this work, each time step is a 3D tensor with $P = 1$ (representing SST) with a grid size of $85 \times 85$.

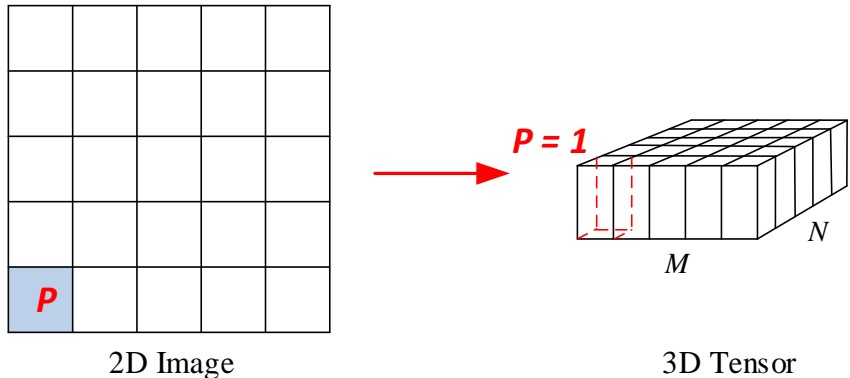

**Figure 1.** Transforming 2D Image into 3D Tensor.

### 2.2. Long Short-Term Memory

In previous studies, LSTM, as a special RNN structure, has been shown to be stable and powerful in the time series prediction model. As shown in Figure 2, LSTM employs two gates to control the content of the cell state $c$: the forget gate, which determines how much of the previous moment's unit state $c_{t-1}$ is retained to the current moment $c_t$; and the input gate, which determines how much of the network's current input $x_t$ is saved in the unit state $c_t$. The LSTM employs an output gate to control how much of the unit state $c_t$ is fed into the LSTM's current output value $h_t$.

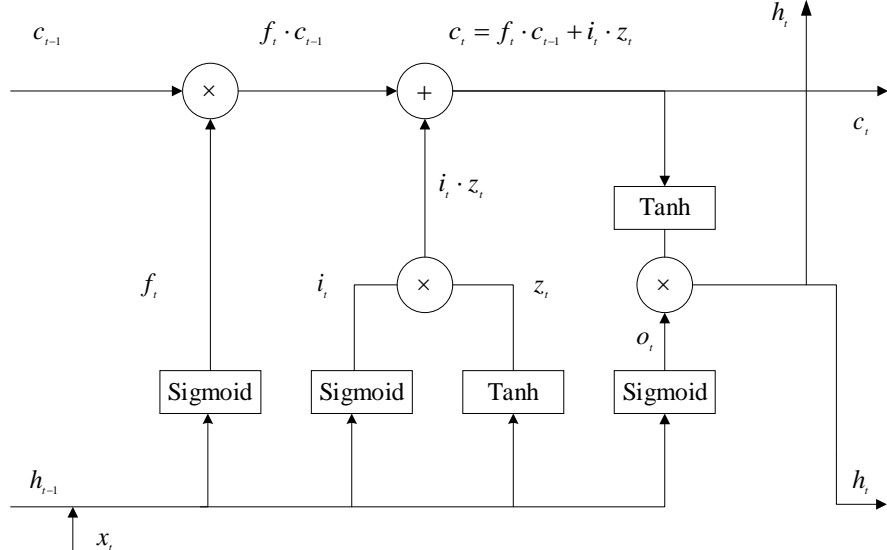

**Figure 2.** LSTM module architecture.

The information state transfer formula of the unit at time $t$ in LSTM is as follows,

$$i_t = \sigma(W_{xi}x_t + W_{hi}h_{t-1} + b_i)$$
$$f_t = \sigma(W_{xf}x_t + W_{hf}h_{t-1} + b_f)$$
$$c_t = f_t \cdot c_{t-1} + i_t \cdot \tanh(W_{xc}x_t + W_{hc}h_{t-1} + b_c) \tag{2}$$
$$o_t = \sigma(W_{xo}x_t + W_{ho}h_{t-1} + b_o)$$
$$h_t = o_t \cdot \tanh(c_t)$$

where $f_t$ is the forget gate processing formula, $i_t$ is the input gate processing formula, $o_t$ is the output gate processing formula, $W$ is the given weight matrix, $\sigma$ is the sigmoid function, and $\cdot$ is the Hadamard product.

To form more complex structures, multiple LSTMs can be stacked and temporally concatenated. Although LSTM has proven to be powerful in dealing with time series problems, the main disadvantage of LSTM when dealing with spatiotemporal data is that the input features must be unrolled into 1D vectors before processing, so all spatial information is lost during processing.

## 3. Materials and Methods

### 3.1. Data

In this study, the reanalysis data used in the established SST model are collected from Copernicus Marine Service (CMEMS). Global ocean reanalyses are homogeneous 3D gridded descriptions of the physical state of the ocean over several decades produced using a numerical ocean model constrained by data assimilation of satellite and in situ observations. The ensemble mean may even provide a more reliable estimate than any individual reanalysis product. Table 1 contains all of the detailed data information used in this experiment. More information can be viewed through the following link: https://data.marine.copernicus.eu/product/GLOBAL_REANALYSIS_PHY_001_031/description (accessed on 12 March 2023).

**Table 1.** Data Sources.

| Input | Time Dimension | Spatial Dimension | Temporal Resolution | Spatial Resolution |
|---|---|---|---|---|
| SST | 2015–2019 | 5°N–26°N, 105°E–126°E | Daily Mean | 0.25° × 0.25° |

### 3.2. Methods

#### 3.2.1. ConvLSTM

Convolutional neural networks and cyclic neural networks are combined to create ConvLSTM. Like LSTM, it can process time series, and like CNN, it can characterize local spatial properties. A more complex architecture can be formed by superimposing multiple ConvLSTM modules to solve the problem of spatiotemporal sequence prediction. Figure 3 depicts the ConvLSTM model's structural layout.

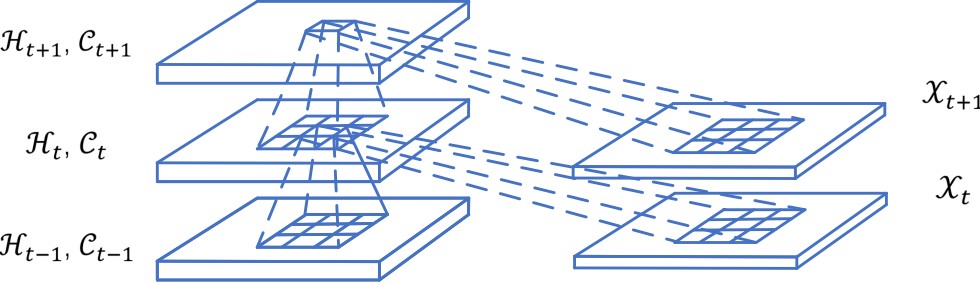

**Figure 3.** ConvLSTM module architecture.

The following is the information state transfer formula for the unit in ConvLSTM at time $t$:

$$
\begin{aligned}
i_t &= \sigma(W_{xi} * \mathcal{X}_t + W_{hi} * \mathcal{H}_{t-1} + W_{ci} \cdot \mathcal{C}_{t-1} + b_i) \\
f_t &= \sigma(W_{xf} * \mathcal{X}_t + W_{hf} * \mathcal{H}_{t-1} + W_{cf} \cdot \mathcal{C}_{t-1} + b_f) \\
\mathcal{C}_t &= f_t \cdot \mathcal{C}_{t-1} + i_t \cdot \tanh(W_{xc} * \mathcal{X}_t + W_{hc} * \mathcal{H}_{t-1} + b_c) \\
o_t &= \sigma(W_{xo} * \mathcal{X}_t + W_{ho} * \mathcal{H}_{t-1} + W_{co} \cdot \mathcal{C}_t + b_o) \\
\mathcal{H}_t &= o_t \cdot \tanh(\mathcal{C}_t)
\end{aligned}
\tag{3}
$$

All of the inputs $\mathcal{X}_1, \ldots, \mathcal{X}_t$, cell outputs $\mathcal{C}_1, \ldots, \mathcal{C}_{t-1}$, hidden state $\mathcal{H}_1, \ldots, \mathcal{H}_t$, and gates $i_t$, $f_t$, $o_t$ in ConvLSTM are 3D tensors in $\mathbb{R}^{P \times M \times N}$, where the first dimension is the number of measurements (for inputs) or feature maps, the last two dimensions are spatial ($M$ rows and $N$ columns), and $*$ denotes the convolution operator and $\cdot$ as before, denotes the Hadamard product.

3.2.2. ST-ConvLSTM

As shown in Figure 4a, the input frame is sent into the first layer of a 4-layer ConvLSTM network, and the future prediction sequence is created in the fourth layer. In this process, hidden states are passed from bottom to top as the information is encoded layer by layer. In this case, as shown by the red and yellow boxes in Figure 4a, the bottom layer will completely ignore what the top layer memorized in the previous time step.

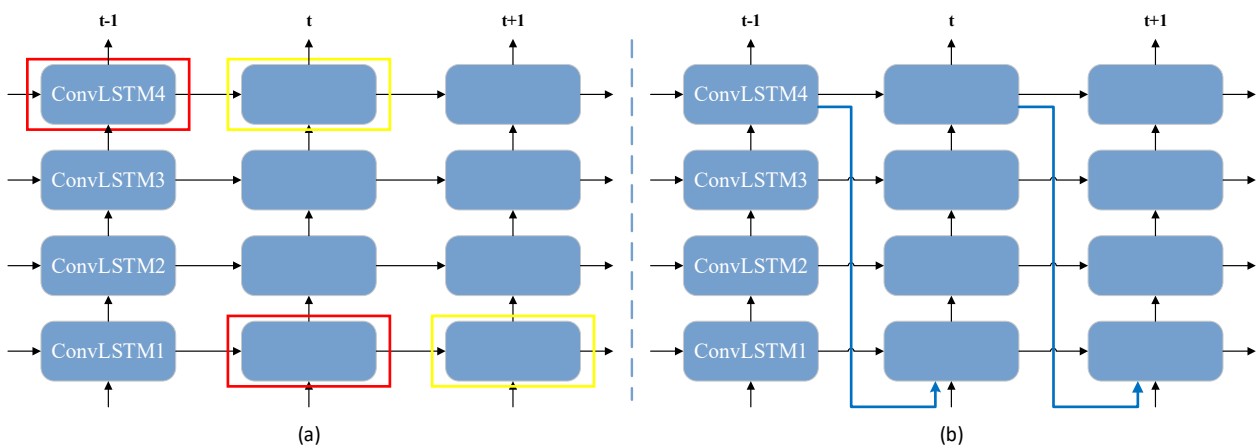

**Figure 4.** (**a**) The conventional ConvLSTM architecture. (**b**) ST-ConvLSTM architecture.

However, if a robust model needs to learn from features at different levels, details in the input sequence should not be lost. In response to the above problems, the model is specially designed by passing the feature information of the fourth-layer ConvLSTM at time $t-1$ to the first-layer ConvLSTM module at time $t$, as highlighted by the blue line in Figure 4b. The information is first transmitted upwards between layers and is transmitted forward as time goes by, and the information of the top layer at the previous moment flows into the bottom layer at this moment for integration, enabling the effective transmission of spatial information.

## 4. Experimental Design

### 4.1. Experimental Environment

All models are trained using the Adam optimizer [54] with a starting learning rate of 0.0001. The training process is stopped after 20,000 iterations. All experiments are implemented in PyTorch [55] and conducted on an NVIDIA 3070 GPU. Other detailed parameter information from the experiment is listed in Table 2.

**Table 2.** Parameters Setting.

| Parameter | Setting |
|---|---|
| Input length | 1 / 3 / 5 / 7 / 15 / 30 |
| Prediction length | 1 / 2 / 4 / 6 / 8 / 10 / 15 |
| Hidden size | 2 / 4 / 8 / 16 / 32 / 64 / 128 |
| Layers | 4 |
| Filter size | $3 \times 3$ |
| Stride | 1 |
| Batch size | 20 |
| Patch size | 5 |
| Test interval | 100 |
| Image size | $85 \times 85$ |
| Image channel | 1 |

## 4.2. Experimental Procedures

In this study, all methods can achieve end-to-end training, and the entire calculation process does not require manual processing but is completely handed over to the deep learning model, from learning the input data feature to obtaining the result. The advantage of end-to-end training is that it reduces the complexity of computational processing. The overall flow of the experimental design is shown in Figure 5.

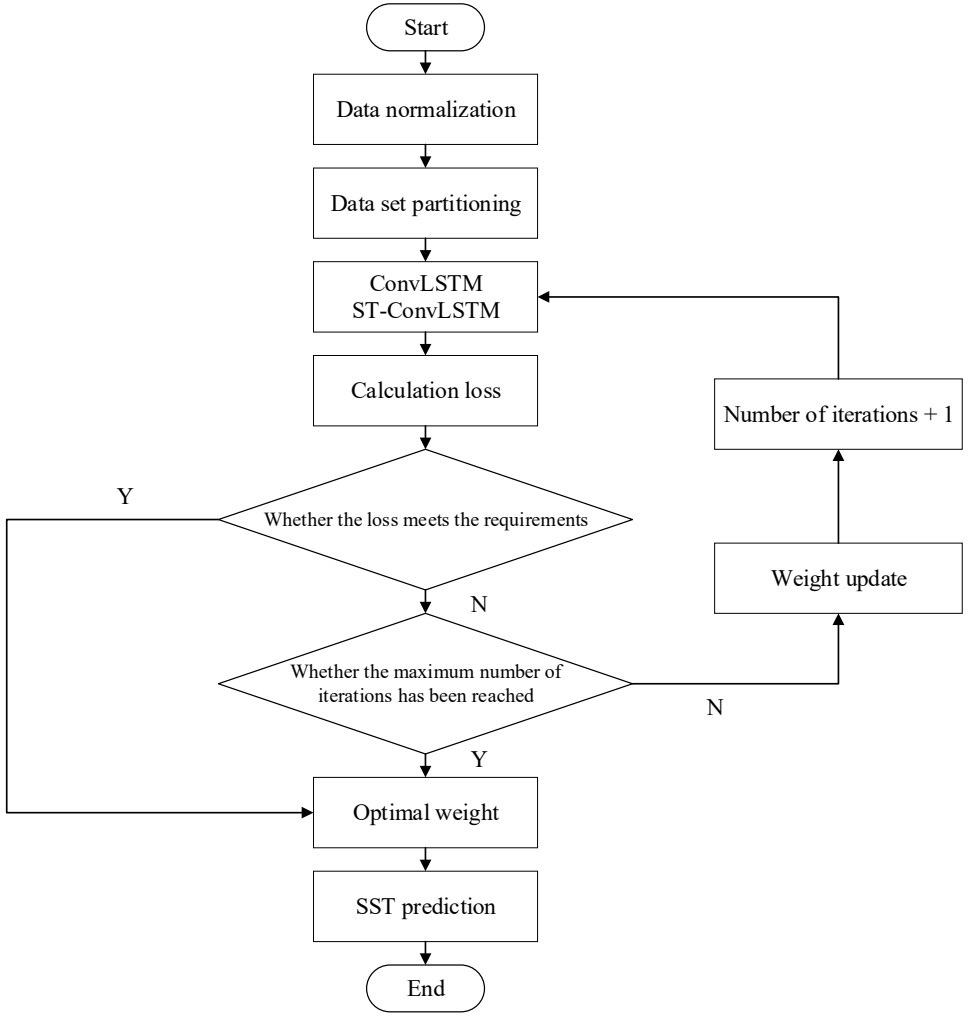

**Figure 5.** Experimental flow chart.

The detailed steps of the SST prediction experiment are as follows.

1. Data preprocessing, using the Numpy library to normalize the input data.
2. Divide the data, using the data from 2015 to 2018 as the training set and the data in 2019 as the validation set.
3. Set a fixed random seed to ensure that each experiment can be reproduced.
4. Model training, using the Adam optimization function to iteratively train the model, and automatically save the optimal weight.
5. Visualize the experimental results and intuitively compare the SST prediction ability of different methods.

*4.3. Metrics*

We use the following three measures to assess the model's performance: root mean square error ($RMSE$), mean absolute error ($MAE$), and coefficient of determination ($R^2$). The following are the calculation algorithms for the above-mentioned three metrics,

$$RMSE = \sqrt{\frac{1}{n}\sum_{i=1}^{n}(\hat{y}_i - y_i)^2} \tag{4}$$

$$MAE = \frac{1}{n}\sum_{i=1}^{n}|\hat{y}_i - y_i| \tag{5}$$

$$R^2 = 1 - \frac{\sum_{i=1}^{n}(\hat{y}_i - y_i)^2}{\sum_{i=1}^{n}(y_i - \overline{y})^2} \tag{6}$$

where $n$ is the total number of test samples, $y_i$, $\hat{y}_i$ and $\overline{y}$ are the true value, the predicted value, and the arithmetic mean of $y_i$, respectively. Note that lower values of $RMSE$ and $MAE$ indicate better agreement between input and prediction, but higher values of $R^2$ indicate more accurate predictions.

## 5. Results

*5.1. Effect of Input Length on SST Prediction Performance*

To verify the influence of the input length on the SST prediction results, when the initial learning rate is 0.0001, the hidden size is 64, the prediction length is 6, and the input lengths are set to 1, 3, 5, 7, 15, and 30, respectively. The influence of input length on the prediction of SST is shown in Table 3, in which the bold font is the optimal result of this group of experiments.

**Table 3.** Effect of input length on SST prediction.

| Input Length | ConvLSTM | | | ST-ConvLSTM | | |
|:---:|:---:|:---:|:---:|:---:|:---:|:---:|
| | *RMSE* | *MAE* | $R^2$ | *RMSE* | *MAE* | $R^2$ |
| 1-d | **0.2559** | **0.1885** | **0.9838** | **0.2673** | **0.2008** | **0.9824** |
| 3-d | 0.2864 | 0.2155 | 0.9798 | 0.2799 | 0.2087 | 0.9807 |
| 5-d | 0.3217 | 0.2372 | 0.9745 | 0.2994 | 0.2187 | 0.9779 |
| 7-d | 0.3280 | 0.2481 | 0.9735 | 0.3019 | 0.2290 | 0.9775 |
| 15-d | 0.3683 | 0.2796 | 0.9671 | 0.3384 | 0.2490 | 0.9722 |
| 30-d | 0.3640 | 0.2734 | 0.9646 | 0.2791 | 0.2069 | 0.9792 |

From the experimental results in Table 3, it can be seen that the two models, ConvLSTM and ST-ConvLSTM, do not have better SST prediction performance when used with longer input lengths. Within the input length range of 1–15, the SST prediction performance of the models gradually decreases as the input length increases. However, at the input length of

30, the SST prediction performance of the model is improved to some extent, but there is still a gap compared to the prediction index obtained at the input length of 1.

The optimal results are obtained with an input length of 1 for both methods when the other conditions are held constant and only the input length is changed. The possible reason for this is that the model fully extracts and learns the feature information contained in the data. If the input length increases, the model cannot fully extract the feature information from the data, and thus the model's SST prediction performance decreases as the input length increases. It is worth mentioning that as the input length increases, the computational effort also increases significantly, instead of achieving better results.

To show more intuitively the comparison of the prediction performance with different parameter settings, we have plotted Figures 6 and 7. From the description in Section 3.2, it is also clear that ST-ConvLSTM is an improved version of ConvLSTM, but from the prediction results of the two methods, ConvLSTM still has an advantage over ST-ConvLSTM when the input length is 1. However, as the input length increases, the prediction performance of ST-ConvLSTM gradually outperforms that of ConvLSTM, which is mainly attributed to ST-ConvLSTM's unique design, which enables the effective transfer of spatial feature information.

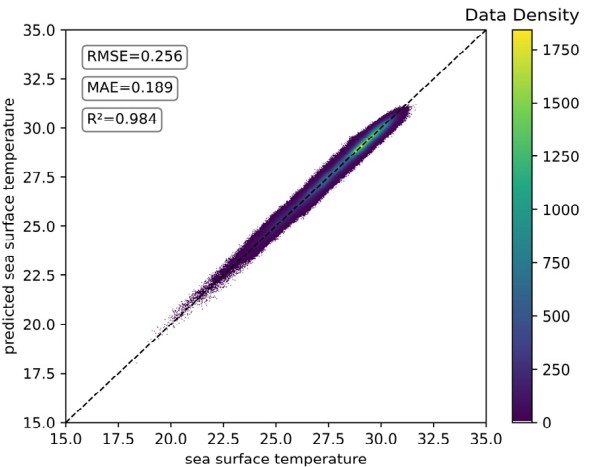 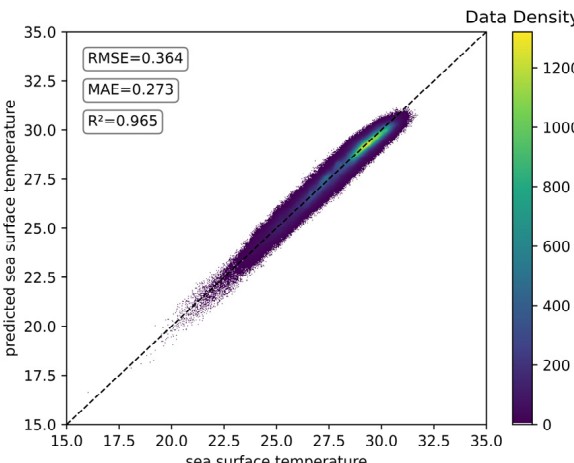

**Figure 6.** Comparison of SST prediction performance using ConvLSTM. On the left, the input length is set to 1, and on the right, it is set to 30.

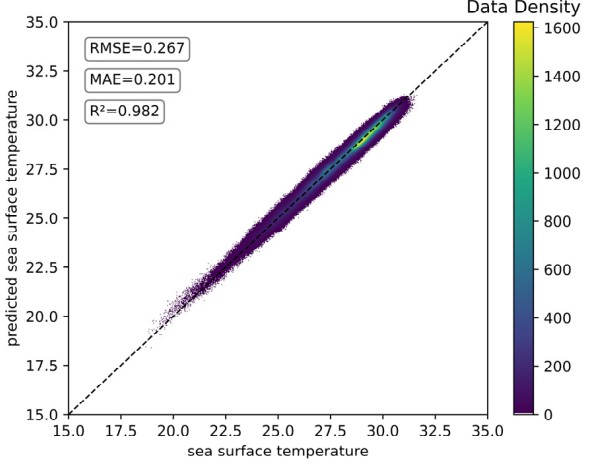 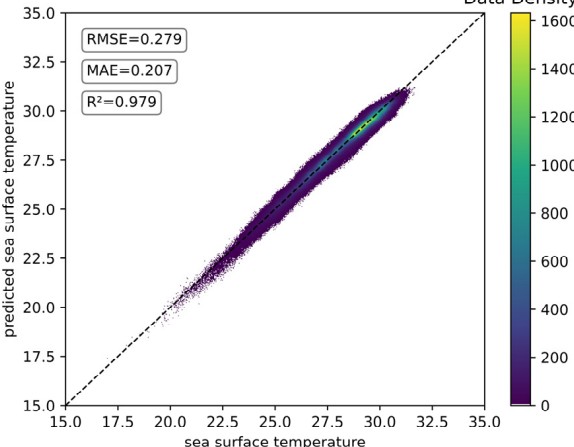

**Figure 7.** Comparison of SST prediction performance using ST-ConvLSTM. On the left, the input length is set to 1, and on the right, it is set to 30.

### 5.2. Effect of Prediction Length on SST Prediction Performance

To verify the influence of prediction length on SST prediction results, according to the analysis of experimental results in Section 5.1, the initial learning rate was set as 0.0001, the hidden size as 64, the input length as 1, and the prediction length as 1, 2, 4, 6, 8, 10 and 15, respectively. The influence of prediction length on the prediction of SST is shown in Table 4, in which the bold font is the optimal result of this group of experiments.

**Table 4.** Effect of prediction length on SST prediction.

| Prediction Length | ConvLSTM | | | ST-ConvLSTM | | |
|:---:|:---:|:---:|:---:|:---:|:---:|:---:|
| | *RMSE* | *MAE* | *$R^2$* | *RMSE* | *MAE* | *$R^2$* |
| 1-d | 0.2463 | 0.1833 | 0.9856 | **0.2195** | **0.1643** | **0.9886** |
| 2-d | **0.2443** | **0.1827** | **0.9859** | 0.2287 | 0.1693 | 0.9877 |
| 4-d | 0.2538 | 0.1895 | 0.9849 | 0.2705 | 0.2029 | 0.9828 |
| 6-d | 0.2559 | 0.1885 | 0.9838 | 0.2673 | 0.2008 | 0.9824 |
| 8-d | 0.3031 | 0.2381 | 0.9775 | 0.2717 | 0.2085 | 0.9819 |
| 10-d | 0.2921 | 0.2202 | 0.9792 | 0.2744 | 0.2065 | 0.9817 |
| 15-d | 0.3055 | 0.2331 | 0.9776 | 0.3272 | 0.2595 | 0.9743 |

It can be seen from Table 4 that with the increase in the prediction length, the sea surface temperature prediction performance of ConvLSTM and ST-ConvLSTM has a gradual decline trend. For ConvLSTM, when the prediction length is 10, the sea surface temperature prediction performance rebounds slightly. For ST-ConvLSTM, when the prediction length is 15, the sea surface temperature prediction performance is much worse than when the prediction length is 10. When other conditions were kept constant and only the prediction length was changed, the ConvLSTM obtained the optimal results at the prediction length of 2, and the ST-ConvLSTM obtained the optimal results at the prediction length of 1.

In order to show more intuitively the comparison of the prediction performance with different parameter settings, we have plotted Figures 8 and 9. Comparing the two models, ST-ConvLSTM does not always outperform ConvLSTM, and ConvLSTM outperforms ST-ConvLSTM in predicting SST at input lengths of 4, 6, and 15 instead. This also means that the overall performance of ST-ConvLSTM is not better than that of ConvLSTM, and specific considerations are needed for SST prediction.

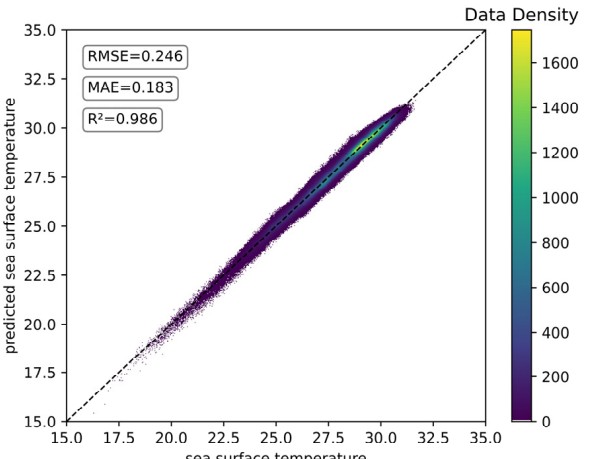 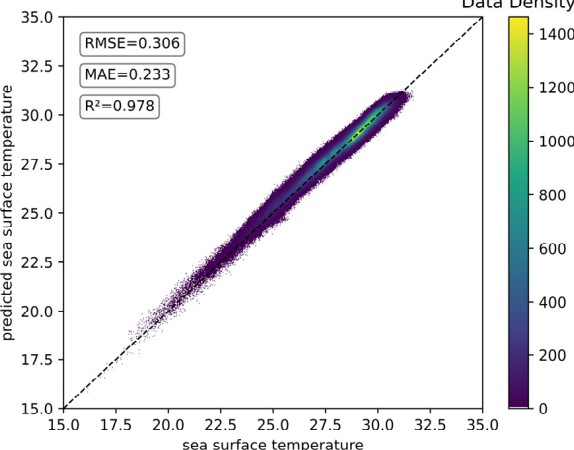

**Figure 8.** Comparison of SST prediction performance using ConvLSTM. On the left, the prediction length is set to 1, and on the right, it is set to 15.

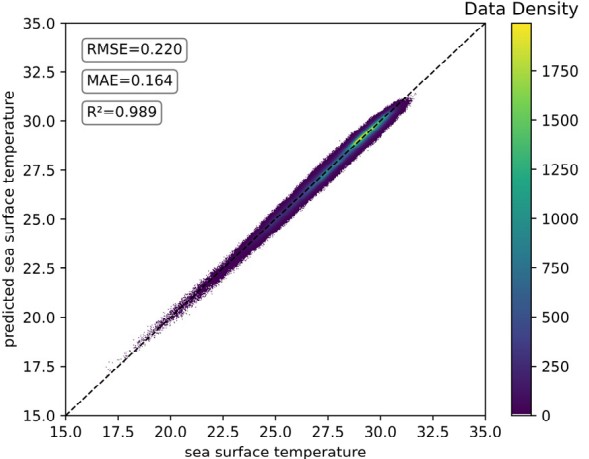
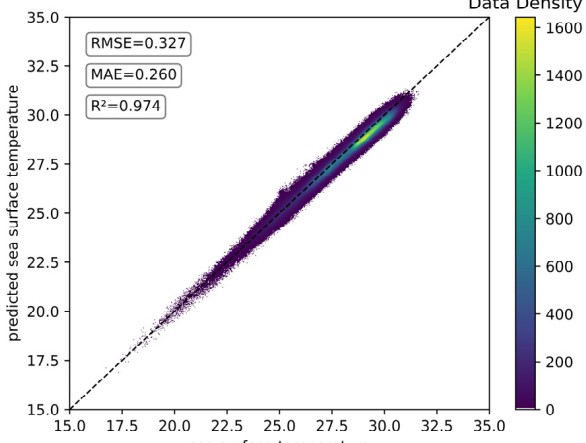

**Figure 9.** Comparison of SST prediction performance using ST-ConvLSTM. On the left, the prediction length is set to 1, and on the right, it is set to 15.

### 5.3. Effect of Hidden Size on SST Prediction Performance

To verify the influence of the hidden layers on the SST prediction results, when the initial learning rate is 0.0001, the input length is 1, and the prediction length is 10. The influence of hidden size on the prediction of SST is shown in Table 5, in which the bold font is the optimal result of this group of experiments.

**Table 5.** Effect of hidden size on SST prediction.

| Hidden Size | ConvLSTM | | | ST-ConvLSTM | | |
| --- | --- | --- | --- | --- | --- | --- |
| | *RMSE* | *MAE* | $R^2$ | *RMSE* | *MAE* | $R^2$ |
| 2 | 4.0448 | 1.8103 | −2.9753 | 3.9653 | 1.7515 | −2.8206 |
| 4 | 3.3343 | 1.4934 | −1.7014 | 3.3333 | 1.4852 | −1.6997 |
| 8 | 2.5523 | 1.1545 | −0.5828 | 2.5377 | 1.1426 | −0.5647 |
| 16 | 1.4760 | 0.7289 | 0.4706 | 1.4820 | 0.7500 | 0.4662 |
| 32 | 0.3670 | 0.2825 | 0.9672 | 0.2953 | 0.2220 | 0.9788 |
| 64 | **0.2921** | **0.2202** | **0.9792** | 0.2744 | 0.2065 | 0.9817 |
| 128 | 0.3226 | 0.2598 | 0.9747 | **0.2459** | **0.1821** | **0.9852** |

From Table 5, we can see that the setting of hidden size has a large impact on the prediction ability of SST. To show more intuitively the comparison of the prediction performance with different parameter settings, we have plotted Figures 10 and 11. From the perspective of ConvLSTM, as the value of the hidden size increases, the prediction performance gradually improves, but it does not mean that the larger the setting, the better the prediction performance. When the hidden size is set to 128, the prediction performance of sea surface temperature starts to decrease. The possible reason for this is that the input feature information is less, and an overly complex network structure has side effects on the prediction of sea surface temperature. From ST-ConvLSTM, as the hidden size increases, the prediction performance gradually improves. This may be due to the unique design of ST-ConvLSTM, which ensures that the feature information is not lost.

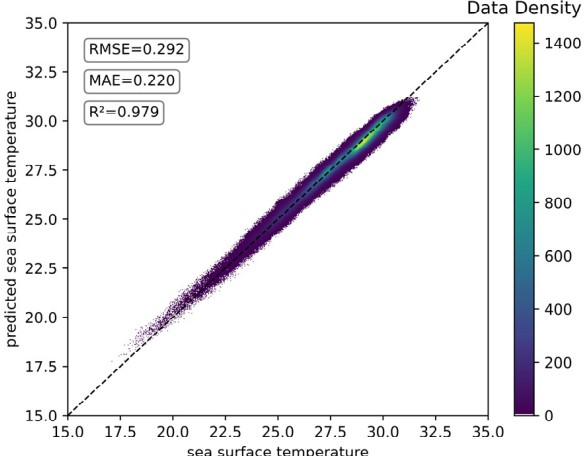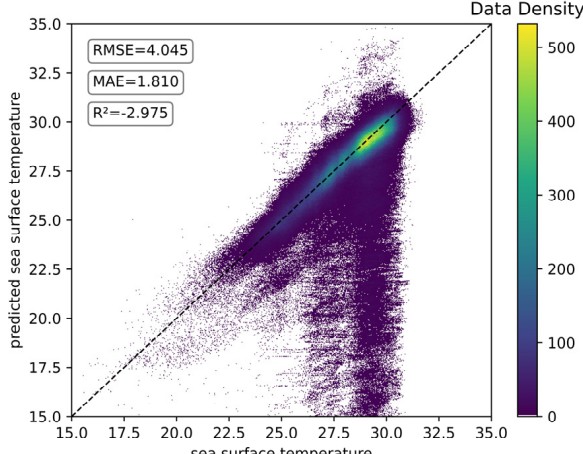

**Figure 10.** Comparison of SST prediction performance using ConvLSTM. On the left, the hidden size is set to 64, and on the right, it is set to 2.

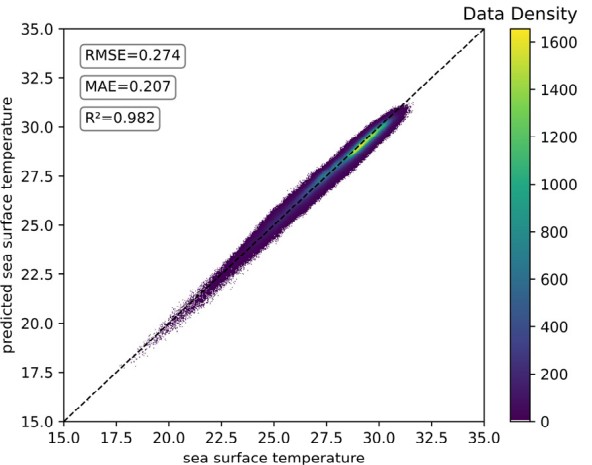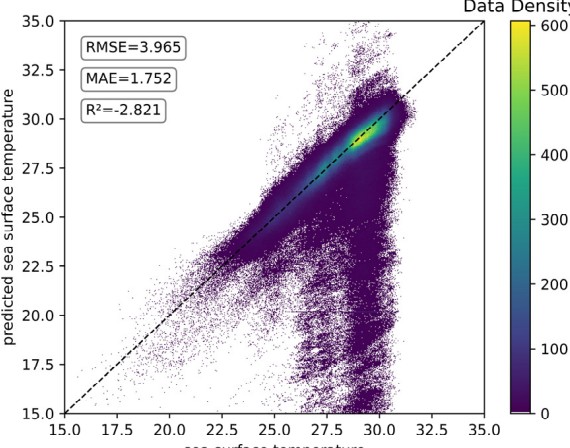

**Figure 11.** Comparison of SST prediction performance using ST-ConvLSTM. On the left, the hidden size is set to 64, and on the right, it is set to 2.

It is worth mentioning that although the prediction performance of SST is better when the hidden size is set to 128 than when it is set to 64, the accompanying computational cost is increased by about 50% and the performance is only improved by about 10%.

### 5.4. Visualization of SST Prediction Performance

Capturing the variability of SST plays an important role in our study and use of the ocean. To better study the SST prediction ability under different environments, we chose March, June, September, and December 2019 from 21 to 30 for testing. After the initial learning rate was set to 0.0001, the hidden size was set to 64, the input length was set to 1, the prediction length was set to 10, and 20,000 rounds of iterative training, the advantages and disadvantages of ConvLSTM and ST-ConvLSTM in predicting SST under different time nodes were analyzed comprehensively by visualizing the difference between the "Ground Truth" and the "Predicted". As shown in Table 6, Figures 12 and 13, where "Ground Truth" represents the true SST value, "Predicted" represents the model predicted SST value, and "Error" represents the difference between the former and the latter.

**Table 6.** The prediction result of SST (Maximum Error and Minimum Error).

| Time Nodes | ConvLSTM | | ST-ConvLSTM | |
|---|---|---|---|---|
| | Max | Min | Max | Min |
| 21 March 2019~30 March 2019 | 1.5590 | −3.3055 | 2.1003 | −2.5679 |
| 21 June 2019~30 June 2019 | 1.4458 | −1.3259 | 1.3666 | −1.2347 |
| 21 September 2019~30 September 2019 | 1.6907 | −1.6543 | 1.3691 | −1.4009 |
| 21 December 2019~30 December 2019 | 1.6345 | −1.9725 | 2.7281 | −2.3366 |

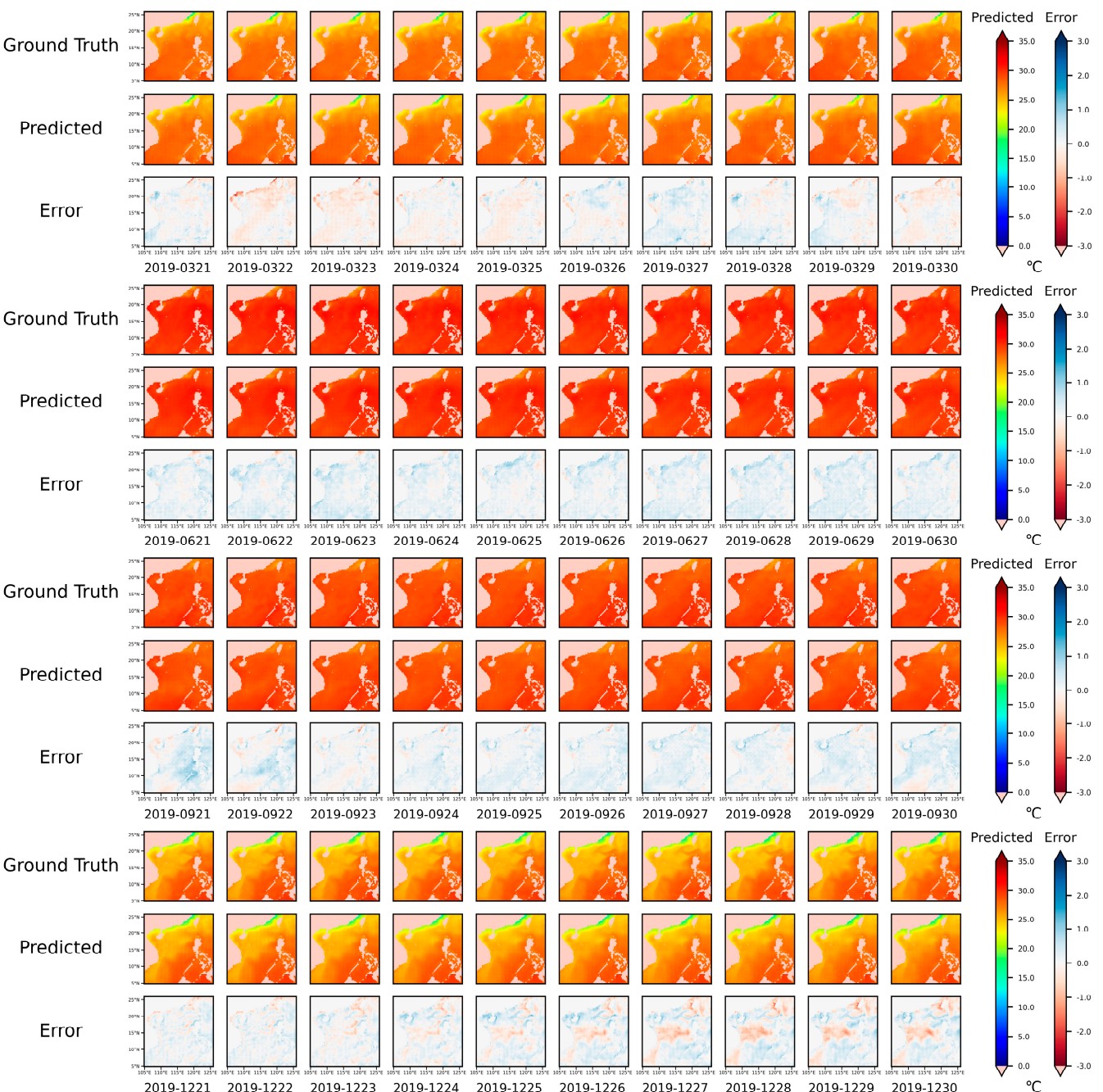

**Figure 12.** Visualization of the ConvLSTM predicted SST for the next 10 days.

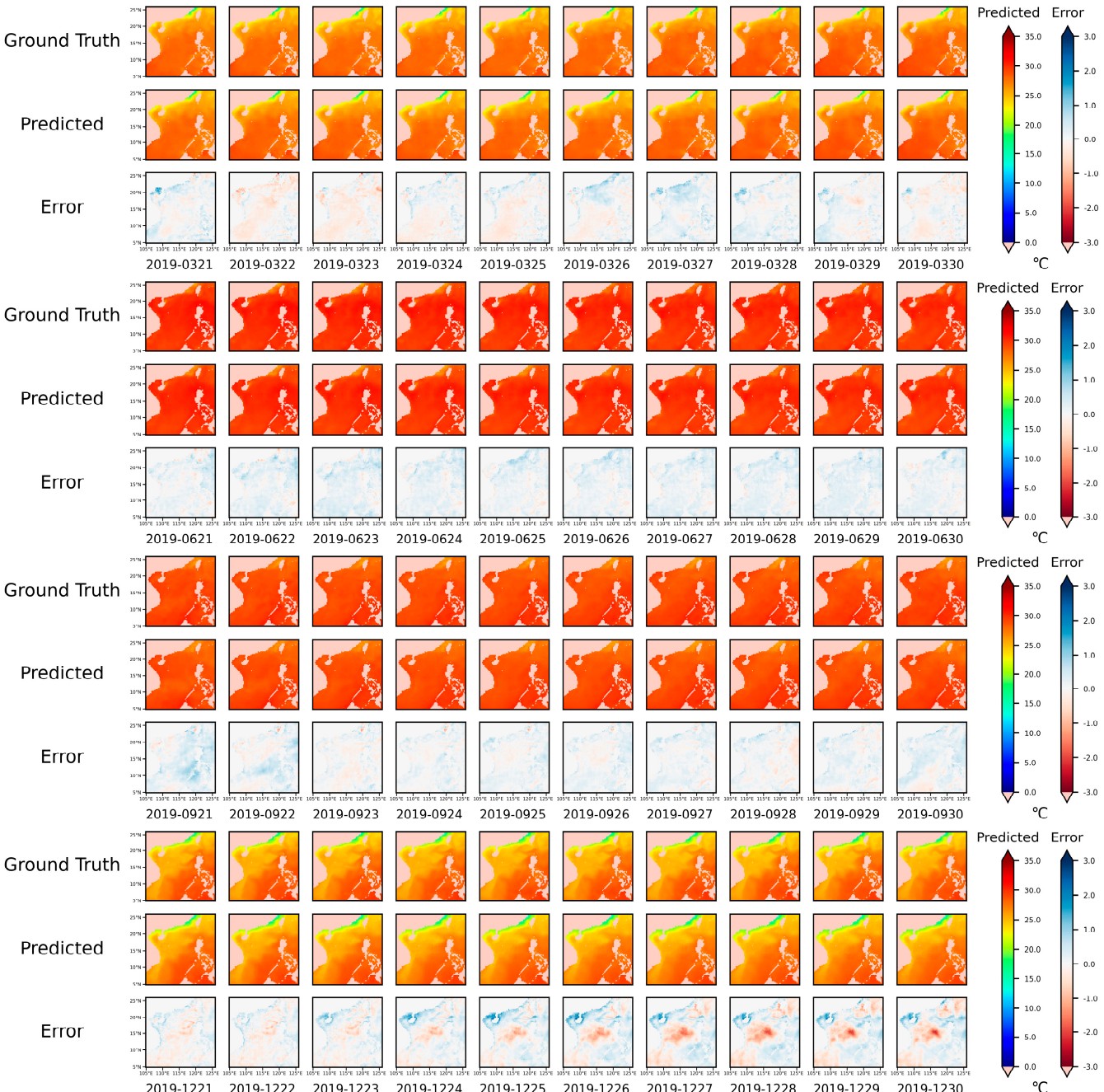

**Figure 13.** Visualization of the ST-ConvLSTM predicted SST for the next 10 days.

The South China Sea is a tropical ocean with high sea surface temperatures, but due to the large latitudinal span and the influence of monsoons and currents, there are differences in the distribution of surface water temperatures between the north and south. From Figures 12 and 13, the smaller the value of "Error", the whiter the image as a whole. When using ConvLSTM and ST-ConvLSTM to predict SST, the predicted values of SST in March and December are higher overall; the predicted values of SST in June and September are lower overall. The highest error of the ConvLSTM prediction reaches −3.3055 in March, and the highest error of the ST-ConvLSTM prediction reaches 2.7281 in December. The possible reason is that the ocean dynamics in the South China Sea are complex, and the mechanism behind the change is not fully learned. We take December as an example, and it is obvious that the ConvLSTM predicts better than the ST-ConvLSTM. So although ST-ConvLSTM is an improved version of ConvLSTM, it is not the case that ST-ConvLSTM

is better than ConvLSTM everywhere, and specific problems need to be analyzed to make reasonable inferences.

## 6. Conclusions

In this study, we used two commonly used spatiotemporal prediction models, ConvLSTM and ST-ConvLSTM, to analyze the difference in the prediction performance of SST by combining different input lengths, prediction lengths, and hidden sizes. The main findings of this study are as follows:

(1) The input length has an effect on SST prediction, but that does not mean that the longer the input length is, the better the prediction performance is. With the same other settings, the two methods, ConvLSTM and ST-ConvLSTM, have the best SST prediction performance when the input length is set to 1. On the whole, the SST prediction performance tends to decrease instead as the input length increases.

(2) The prediction length has an effect on SST prediction. When other parameters are kept constant and only the prediction length is changed, ConvLSTM gets the optimal result when the prediction length is set to 2 and ST-ConvLSTM gets the optimal result when the prediction length is set to 1. The SST prediction performance of ConvLSTM and ST-ConvLSTM tends to decrease gradually as the prediction length increases.

(3) The setting of the hidden size has a large impact on the prediction ability. For ConvLSTM, the prediction performance first gradually improves with the increase in the hidden size value, and the improvement is larger, and then the SST prediction performance starts to decrease when the hidden size is set to 128. For ST-ConvLSTM, the prediction performance gradually improves as the hidden size increases, and the prediction performance of SST is better when the hidden size is set to 128 than when it is set to 64, but then the computational cost increases by about 50% and the performance only improves by about 10%.

Deep learning methods have achieved good results in SST prediction. However, there are some drawbacks: (1) It is like a "black box", and the inference mechanism between model input and output is not clear. (2) These methods rely too much on the size of the input training data, and the model prediction may be poor in the case of a small training set. In our future work, we will focus on model interpretability, model lightweighting, and few-shot learning to make breakthroughs.

**Author Contributions:** Conceptualization, P.H. and S.L.; methodology, P.H.; software, P.H.; validation, S.L., Y.G. and J.S.; formal analysis, P.H.; investigation, J.S.; resources, P.H.; data curation, P.H.; writing—original draft preparation, P.H.; writing—review and editing, J.S.; visualization, P.H.; supervision, S.L.; project administration, S.L.; funding acquisition, J.S. All authors have read and agreed to the published version of the manuscript.

**Funding:** This research was funded by the National Natural Science Foundation of China, grant numbers 41830533 and 41876003.

**Data Availability Statement:** For more information, please refer to the website: https://data.marine.copernicus.eu/product/GLOBAL_REANALYSIS_PHY_001_031/description (accessed on 12 March 2023).

**Conflicts of Interest:** The authors declare no conflict of interest.

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
