# Peer review of "Prediction of Sea Surface Temperature in the South China Sea Based on Deep Learning"

_remotesensing, doi:10.3390/rs15061656_

Round 1

Reviewer 1 Report

In this paper, the authors evaluated how the setting of certain parameters influences the performance of two popular deep-learning models, ConvLSTM and ST-ConvLSTM, in SST prediction. Their results could be helpful in the model design and setting for later researchers. Before the manuscript becomes acceptable, a moderate revision should be made and some concerns should be addressed.

1. The process of training is a critical factor which can decide the performance of the model to a great extent besides model designment. Usually we take as much as able data to train the model, in order to involve as many as the features of the phenomenon in different spatiotemporal scales. My major concern is: as the dataset used in this paper (CMEMS Global Ocean Ensemble Physics Reanalysis) has a temporal length of about 28 years, why the authors only take a 5-years-long segment from it? Did they study how the selection of temporal length of the training data impacts the model performance or not? It would be better illuminated by a set of experiments with different temporal lengths of training data.

2. Still the issue of training data. The authors used CMEMS 2015-2018 data to train their model. However, the climatological oscillating period of SST is obviously longer than 4 years (e.g. ENSO). It is neither clear yet that the data segment is at what phase of the period, nor the influence of the climatological variation on the model performance.

3. In this paper the authors trained the model with CMEMS Global Ocean Ensemble Physics Reanalysis SST data. Actually, the remote sensing SST are more extensively used in SST prediction. Since the assimilation of AVISO remote sensing SST data into the CMEMS reanalysis data, these two data should present similar spatiotemporal features. It’s suggested that the authors could check the prediction ability of the model from remote sensing data, and they might consider to train the model with remote sensing data in their future work.

Author Response

We would like to thank you for your constructive comments concerning our article (Manuscript ID: remotesensing-2239467). These comments are all valuable and helpful for improving our article. All the authors have seriously discussed all these comments. According to the comments, we have tried our best to modify our manuscript to satisfy the requirements of this journal and readers. In this revised version, changes to our manuscript within the document were all highlighted by using red colored text.

Answer:

Thank you very much for your valuable comments.

  1. The process of training is a critical factor which can decide the performance of the model to a great extent besides model designment. Usually we take as much as able data to train the model, in order to involve as many as the features of the phenomenon in different spatiotemporal scales. My major concern is: as the dataset used in this paper (CMEMS Global Ocean Ensemble Physics Reanalysis) has a temporal length of about 28 years, why the authors only take a 5-years-long segment from it? Did they study how the selection of temporal length of the training data impacts the model performance or not? It would be better illuminated by a set of experiments with different temporal lengths of training data.

Answer:

Thank you for your good suggestion. As you said, a larger data set is generally chosen because it is known that using a larger data set tends to work better because more feature information is included. With this conclusion in mind, we did not use a larger dataset for two reasons:

(1) Section 4.1 gives information on the specific parameters and equipment we used, and we did the best we could within our capabilities, and we may not have been able to meet the normal computational processing if the data volume was any larger.

(2) From the metrics in Table 3 Table 4 Table 5, the prediction performance has already reached a high level, and if we keep expanding the size of the training set, we may get a slight improvement in performance, but the price brought is a doubling of the computational cost, which is an approach that outweighs the loss.

  1. Still the issue of training data. The authors used CMEMS 2015-2018 data to train their model. However, the climatological oscillating period of SST is obviously longer than 4 years (e.g. ENSO). It is neither clear yet that the data segment is at what phase of the period, nor the influence of the climatological variation on the model performance.

Answer:

Thank you for your good suggestion. Your question is really too valuable for our future research. As we answered your first question, we explained the reason for using only a partial dataset. Regarding the problem you mentioned, it is the same problem that deep learning methods have been facing since their hot application, that is, the interpretability problem, which is like a black box due to poor interpretability.

At present, even if we use large-scale data, we still cannot reasonably solve this interpretability problem. Our group is trying to combine physically driven and deep learning methods for this field. The theoretical basis of this approach lies in the general approximation theorem for neural networks, which allows them to approximate arbitrary continuous functions very accurately. The output of a neural network is a continuous, differentiable function, so that the exact derivative of the network output with respect to each input variable can be automatically computed using the BackProp algorithm, allowing each term of the partial differential equation to be computed exactly at the configured sampling points in the region. The algorithm is to embed the residual terms of the partial differential equations thus computed after automatic differentiation as regular term constraints into the problem-specific loss function, which is a soft penalty constraint that can be applied to any complex spatiotemporal region and any type of boundary conditions. After the network is trained, the algorithm will output a function that approximates the solution of the partial differential equation.

(1) Compared with traditional numerical methods, the algorithm is a gridless method, so that the network solutions so output can be well adapted to different spatial and temporal resolutions. Since partial differential equation constraints can greatly reduce the complexity of the hypothesis space, such a network output can well approximate the solution structure of partial differential equations.

(2) Compared to classical neural networks, such a model has relatively good (physical) interpretability as well as generalization ability due to the embedded specific physical prior information.

In addition, as mentioned before, using large datasets brings an exponential increase in computational cost, which we think is a means to lose more than gain if it only improves the performance marginally. Again, thank you for your valuable comments and we hope this will explain to you why. Also, you have pointed us in the direction of future research. In our future work, we will focus on model interpretability, model lightweight, and few-shot learning to make breakthroughs.

  1. In this paper the authors trained the model with CMEMS Global Ocean Ensemble Physics Reanalysis SST data. Actually, the remote sensing SST are more extensively used in SST prediction. Since the assimilation of AVISO remote sensing SST data into the CMEMS reanalysis data, these two data should present similar spatiotemporal features. It’s suggested that the authors could check the prediction ability of the model from remote sensing data, and they might consider to train the model with remote sensing data in their future work.

Answer:

Thank you for your good suggestion. Regarding this issue, we consider it from the following aspects: (1) When training and prediction, the data involved in prediction is not included in the already trained, and the experimental results also show that the method is effective. (2) The use of processed reanalysis data eliminates some of our data pre-processing processes. (3) The reanalysis data can also reflect the information of data characteristics to a large extent. (4) We have used these 12+14+14 sets of experiments to provide a comprehensive measure of the effect of different parameter settings of different methods on the prediction performance, and we have achieved our original purpose.

Regarding this suggestion of yours, we will take it into consideration in our next research work, and it is always our pursuit to be closer to the reality, and we will step by step to conduct the related research. We hope that our explanation will allay your concerns and thank you again for your valuable comments, which have made us gain a lot.

Reviewer 2 Report

The SST prediction performance of ConvLSTM and ST-ConvLSTM were studied with different input lengths, prediction lengths, and hidden sizes. This is a hot topic. Several groups of experiments have been done. However, some parts need to be improved before the manuscript can be accepted.

1. Ln 91: the subtitle should be reformulated. It is recommended to use "deep learning".

2. Section 3.1: the title should be changed to "Data". The introduction of data in this part is not clear enough. CMEMS has many datasets.

3. Table 3, 4, and 5: where is the "bold font"? All the three tables have the same captions. Please reformulate the captions and give more details. 

4. Ln 256: "Table 3" should be corrected to "Table 4".

5. The English writing should be strengthened. Some sentences are too long, for example Ln 262-266. Some sentences are strange, for example Ln 300-303. Some sentences are too colloquial. 

6. The presentation of results should be improved. The scatter plots in Figure 8-9 do not tell the difference clearly. Time series can be considered.

7. The presentation includes many subplots in Figure 12-13 which is hard to read, discuss, or focus about anything specific. The authors do not need to present everything as this is not a project report. I suggest you select a few typical subplots and discuss them in detail.

8. The analysis on Figure 12-13 should be further strengthened. The presentation in Ln 339-352 is not so clear.

9. "Discussion" part should be added, including comparison with the previous studies and the prospects of both neural networks.

Author Response

We would like to thank you for your constructive comments concerning our article (Manuscript ID: remotesensing-2239467). These comments are all valuable and helpful for improving our article. All the authors have seriously discussed all these comments. According to the comments, we have tried our best to modify our manuscript to satisfy the requirements of this journal and readers. In this revised version, changes to our manuscript within the document were all highlighted by using red colored text.

  1. Ln 91: the subtitle should be reformulated. It is recommended to use "deep learning".

Answer:

Thank you for your good suggestion. We have revised the subtitle.

  1. Section 3.1: the title should be changed to "Data". The introduction of data in this part is not clear enough. CMEMS has many datasets.

Answer:

Thank you very much for your valuable comments. As you mentioned, CMEMS does have a lot of data sets, but we have given specific links to the data sets we used. To more intuitively represent the core data information, we have given Table 1, so You will have a more intuitive understanding of the data.

  1. Table 3, 4, and 5: where is the "bold font"? All the three tables have the same captions. Please reformulate the captions and give more details.

Answer:

Thank you for your good suggestion. We have carefully checked the information in the table, and we have neglected the bold mark. Now we have made changes to the table, which will show the performance of different models more intuitively.

  1. Ln 256: "Table 3" should be corrected to "Table 4".

Answer:

Indeed, we made a mistake here, thank you for pointing this out, and we have made corrections in the original text. Thank you again for your seriousness and responsibility.

  1. The English writing should be strengthened. Some sentences are too long, for example Ln 262-266. Some sentences are strange, for example Ln 300-303. Some sentences are too colloquial.

Answer:

Thank you for your good suggestion. In response to this problem, we have repeatedly revised the language aspect. Thank you for pointing out our problem. We have also carefully revised the two places you mentioned.

  1. The presentation of results should be improved. The scatter plots in Figure 8-9 do not tell the difference clearly. Time series can be considered.

Answer:

Thank you for your good suggestion. Regarding your question, I think I did not explain it clearly in the original article, so I will describe it in detail. Judging from the three indicators alone, the effect does not seem to be very different, but we have given the color bar on the right, which means data density. It can be clearly seen that the scale values are different. I wonder if this explanation can dispel your concerns?

  1. The presentation includes many subplots in Figure 12-13 which is hard to read, discuss, or focus about anything specific. The authors do not need to present everything as this is not a project report. I suggest you select a few typical subplots and discuss them in detail.

Answer:

Thank you very much for your valuable comments. Regarding the question you mentioned, I think I did not describe it clearly. We comprehensively analyze the comparison of the predictive performance of these two methods under different settings throughout the paper. Cold numbers cannot give people an intuitive expression, so we consider doing a visualization so that we can clearly see the direct differences between different models. We have given the color bar. The meaning Error is the direct difference between the prediction and the actual data. Different differences represent different colors and can be clearly displayed. It is not convincing to compare a group alone, because there may be a certain chance, so we took 10 days to do a visualization at intervals. Thank you for your seriousness and responsibility. Can my explanation dispel your doubts?

  1. The analysis on Figure 12-13 should be further strengthened. The presentation in Ln 339-352 is not so clear.

Answer:

Thank you very much for your valuable comments. Just like we answered your seventh question, is it clearer to you? The title of Chapter 5.4 is visualization. We added Table 6, but for visualization, we then gave Figure 12 and Figure 13. ‘Ground Truth’ represents the input value, ‘Predicted’ represents the predicted value, and ‘Error’ represents the error between the two. It is intuitively displayed through the difference in color bar, making it clear at a glance. It is through visualization that we can clearly understand that the two models have their own advantages, and specific issues must be considered in practical applications. The analysis in 5.4 is actually a short-answer summary of 5.1, 5.2, and 5.3. They complement each other and take care of each other.

  1. "Discussion" part should be added, including comparison with the previous studies and the prospects of both neural networks.

Answer:

Thank you very much for your valuable comments. We thought about it carefully. The fifth chapter of the article is not only the display of the experimental results but also the discussion of the experimental results. Coupled with our responses to your questions 7 and 8, I think it should be able to dispel your doubts. In our future work, we will focus on model interpretability, model lightweight, and few-shot learning to make breakthroughs. Thank you again for your many valuable comments on our work. We have marked the specific revisions in red in the article, please refer to them.

Reviewer 3 Report

The study titled "Prediction of Sea Surface Temperature in the South China Sea Based on Deep Learning" is a good effort to predict the sea surface temperature in the South China Sea using deep learning. This authors compared the performance of two spatiotemporal prediction models, ConvLSTM and ST-ConvLSTM, in predicting SST. More details on the following need to be provided:

1. What are the two spatiotemporal prediction models used in this study?

2. What were the parameters that were varied to analyze the performance of the models?

3. What was the effect of the input length on SST prediction performance for both ConvLSTM and ST-ConvLSTM?

4. What was the optimal prediction length for ConvLSTM and ST-ConvLSTM, and how did the SST prediction performance change as the prediction length increased?

5. What was the impact of hidden layer size on prediction ability for ConvLSTM and ST-ConvLSTM?

6. What are some of the drawbacks of deep learning methods in SST prediction, as discussed in this study?

7. What are the authors' plans for future work, based on the findings of this study?

Further, the authors should comment on the use of hybrid dynamical+ deep learning models for Earth system science. There have been some attempts recently on the same, some of them are listed below:

Hwang, J., Orenstein, P., Cohen, J., Pfeiffer, K. and Mackey, L., 2019, July. Improving subseasonal forecasting in the western US with machine learning. In Proceedings of the 25th ACM SIGKDD International Conference on Knowledge Discovery & Data Mining (pp. 2325-2335).

Singh Manmeet, SB Vaisakh, Acharya Nachiketa, Grover Aditya, Rao Suryachandra A., Kumar Bipin, Yang Zong-Liang, Niyogi Dev. Short-range forecasts of global precipitation using deep learning-augmented numerical weather prediction. NeurIPS 2022 Workshop on Tackling Climate Change with Machine Learning

Author Response

We would like to thank you for your constructive comments concerning our article (Manuscript ID: remotesensing-2239467). These comments are all valuable and helpful for improving our article. All the authors have seriously discussed about all these comments. According to the comments, we have tried our best to modify our manuscript to satisfy with the requirements of this journal and readers. In this revised version, changes to our manuscript within the document were all highlighted by using red colored text.

  1. What are the two spatiotemporal prediction models used in this study?

Answer:

Thank you very much for your valuable comments. In this study, the two models we use are ConvLSTM and ST-ConvLSTM. We give specific references as Nos. 47 and 48, respectively.

  1. What were the parameters that were varied to analyze the performance of the models?

Answer:

Thank you very much for your valuable comments. We comprehensively measure the impact of different methods and parameters on the predictive ability of SST by setting different input lengths, prediction lengths, and hidden size in the network.

  1. What was the effect of the input length on SST prediction performance for both ConvLSTM and ST-ConvLSTM?

Answer:

Thank you very much for your valuable comments.

To verify the influence of the input length on the SST prediction results, when the initial learning rate is 0.0001, the hidden size is 64, the prediction length is 6, and the input lengths are set to 1, 3, 5, 7, 15, and 30, respectively, to discuss the effect of input length on SST prediction performance. The influence of input length on the prediction of SST is shown in Table 3, in which the bold font is the optimal result of this group of experiments.

The experimental results show that: the input length has an impact on the prediction results of SST, but it does not mean that the longer the input length the better the prediction performance. ConvLSTM and ST-ConvLSTM have the best prediction performance when the input length is set to 1, and the prediction performance gradually decreases as the input length increases.

  1. What was the optimal prediction length for ConvLSTM and ST-ConvLSTM, and how did the SST prediction performance change as the prediction length increased?

Answer:

Thank you very much for your valuable comments.

To verify the influence of prediction length on SST prediction results, according to the analysis of experimental results in Section 3.1, the initial learning rate was set as 0.0001, the hidden size as 64, the input length as 1, and the output length as 1, 2, 4, 6, 8, 10 and 15, respectively, to discuss the influence of output length on SST prediction performance. The influence of prediction length on the prediction of SST is shown in Table 4, in which the bold font is the optimal result of this group of experiments.

The experimental results show that: Output length affects SST prediction. As the output length increases, the prediction performance gradually decreases. When other parameters are kept constant and only the output length is changed, the ConvLSTM gets the best result when the output length is set to 2, and the ST-ConvLSTM gets the best result when the output length is set to 1.

  1. What was the impact of hidden layer size on prediction ability for ConvLSTM and ST-ConvLSTM?

Answer:

Thank you very much for your valuable comments.

To verify the influence of the hidden layers on the SST prediction results, when the initial learning rate is 0.0001, the input length is 1, and the prediction length is 10, discuss the effect of layers on SST prediction performance. The influence of hidden size on the prediction of SST is shown in Table 5, in which the bold font is the optimal result of this group of experiments.

The experimental results show that: The setting of the hidden size has a great influence on the prediction ability of the sea surface temperature, but the hidden size cannot be set blindly. For ST-ConvLSTM, although the prediction performance of SST is better when the hidden size is set to 128 than when it is set to 64, the consequent computational cost increases by about 50%, and the performance only improve by about 10%.

  1. What are some of the drawbacks of deep learning methods in SST prediction, as discussed in this study?

Answer:

Thank you very much for your valuable comments. Deep learning methods have achieved good results in SST prediction. However, there are some drawbacks: (1) It is like a "black box", and the inference mechanism between model input and output is not clear. (2) These methods rely too much on the size of the input training data, and the model prediction may be poor in the case of a small training set size.

  1. What are the authors' plans for future work, based on the findings of this study?

Answer:

Thank you very much for your valuable comments. In our future work, we will focus on model interpretability, model lightweight, and few-shot learning to make breakthroughs.

Further, the authors should comment on the use of hybrid dynamical+ deep learning models for Earth system science. There have been some attempts recently on the same, some of them are listed below:

Hwang, J., Orenstein, P., Cohen, J., Pfeiffer, K. and Mackey, L., 2019, July. Improving subseasonal forecasting in the western US with machine learning. In Proceedings of the 25th ACM SIGKDD International Conference on Knowledge Discovery & Data Mining (pp. 2325-2335).

Singh Manmeet, SB Vaisakh, Acharya Nachiketa, Grover Aditya, Rao Suryachandra A., Kumar Bipin, Yang Zong-Liang, Niyogi Dev. Short-range forecasts of global precipitation using deep learning-augmented numerical weather prediction. NeurIPS 2022 Workshop on Tackling Climate Change with Machine Learning

Answer:

Thank you for your good suggestion. Both articles you propose are interesting studies. There are two types of combinations of traditional methods and deep learning methods. One is to use the data of traditional methods and then use deep learning methods to predict and analyze; the other is physics-driven deep learning, which is based on prior formulas. Our work belongs to the first category. The combination of traditional methods and deep learning methods may be a research hotspot in the future, and may have the following potential advantages: (1) Compared with traditional numerical methods, this algorithm is a gridless method, so the output of the network solution can be well adapted to different spatial and temporal resolutions, which can greatly reduce the complexity of the hypothesis space. (2) Compared with the classical neural network, such a model has relatively good (physical) interpretability and generalization ability due to the embedded specific physical prior information. As we said in our future work, we will focus on model interpretability, model lightweight, and few-shot learning to make breakthroughs. The specific advantages of the combination of traditional methods and deep learning methods, and how much advantage they have, still need further research.

Round 2

Reviewer 2 Report

The authors have included most of my comments, but some comments are left unanswered. 

Table 3, 4, 5, and 6 almost have the same captions. Please reformulate the captions and give more details.

The English writing should be strengthened. Some sentences are too colloquial, for example "the more...the more...". Please do not just reformulate the two examples I have pointed out. Please polish the Language throughout.

The analysis and discussion of results are still weak.

Author Response

We would like to thank you for your constructive comments concerning our article (Manuscript ID: remotesensing-2239467). These comments are all valuable and helpful for improving our article. All the authors have seriously discussed all these comments. According to the comments, we have tried our best to modify our manuscript to satisfy the requirements of this journal and readers. In this revised version, changes to our manuscript within the document were all highlighted by using red colored text.

  1. Table 3, 4, 5, and 6 almost have the same captions. Please reformulate the captions and give more details.

Answer:

Thank you for your good suggestion. We overlooked this issue, and we have added and revised it in this version.

  1. The English writing should be strengthened. Some sentences are too colloquial, for example "the more...the more...". Please do not just reformulate the two examples I have pointed out. Please polish the Language throughout.

Answer:

Thank you for your good suggestion. Regarding this issue, we have revised and polished the full text, and the modified places are marked in red. We have not shown them here one by one. Please check the original text, we have made detailed marks. Thank you again for controlling the quality of our articles.

  1. The analysis and discussion of results are still weak.

Answer:

Thank you for your good suggestion. Regarding this issue, we have made revisions based on the revision of the first edition, supplemented and introduced how to understand some pictures. Our point is that a good graph is a good annotation. We have made appropriate adjustments and explanations based on your suggestions. Hope this version of the revision clears up your doubts. If there is something that is not satisfactory, we also hope that you can point it out specifically, and we will continue to modify it. Thank you very much for putting forward so many valuable opinions on our article. After carefully revising according to your opinions, the quality of the article has been qualitatively improved.
